# Off-Hours versus Regular-Hours Implantation of Peripheral Venoarterial Extracorporeal Membrane Oxygenation in Patients with Cardiogenic Shock

**DOI:** 10.3390/jcm12051875

**Published:** 2023-02-27

**Authors:** Roberto Gómez-Sánchez, Jorge García-Carreño, Jorge Martínez-Solano, Iago Sousa-Casasnovas, Miriam Juárez-Fernández, Carolina Devesa-Cordero, Ricardo Sanz-Ruiz, Enrique Gutiérrez-Ibañes, Jaime Elízaga, Francisco Fernández-Avilés, Manuel Martínez-Sellés

**Affiliations:** 1Department of Cardiology, Instituto de Investigación Sanitaria Gregorio Marañon (IiSGM), Hospital General Universitario Gregorio Marañón, 28007 Madrid, Spain; 2Centro de Investigación Biomédica en Red-Enfermedades Cardiovasculares (CIBERCV), Instituto de Salud Carlos III, 28029 Madrid, Spain; 3Faculty of Medicine, Universidad Complutense de Madrid, 28040 Madrid, Spain; 4Faculty of Biomedical and Health Sciences, Universidad Europea, 28670 Madrid, Spain; 5Department of Cardiology, Hospital General Universitario Gregorio Marañón, Calle Doctor Esquerdo, 46, 28007 Madrid, Spain

**Keywords:** extracorporeal membrane oxygenation, off-hours, regular-hours, cardiogenic shock, percutaneous cannulation, prognosis

## Abstract

Background. The “weekend effect” has been associated with worse clinical outcomes. Our aim was to compare off-hours vs. regular-hours peripheral venoarterial extracorporeal membrane oxygenation (VA-ECMO) in cardiogenic shock patients. Methods. We analyzed in-hospital and 90-day mortality among 147 consecutive patients treated with percutaneous VA-ECMO for medical reasons between July 1, 2013, and September 30, 2022, during regular-hours (weekdays 8:00 a.m.–10:00 p.m.) and off-hours (weekdays 10:01 p.m.–7:59 a.m., weekends, and holidays). Results. The median patient age was 56 years (interquartile range [IQR] 49–64 years) and 112 (72.6%) were men. The median lactate level was 9.6 mmol/L (IQR 6.2–14.8 mmol/L) and 136 patients (92.5%) had a Society for Cardiovascular Angiography and Interventions (SCAI) stage D or E. Cannulation was performed off-hours in 67 patients (45.6%). In-hospital mortality was similar in off-hours and regular hours (55.2% vs. 56.3%, *p* = 0.901), as was the 90-day mortality (58.2% vs. 57.5%, *p* = 0.963), length of hospital stay (31 days [IQR 16–65.8 days] vs. 32 days [IQR 18–63 days], *p* = 0.979), and VA-ECMO related complications (77.6% vs. 70.0%, *p* = 0.305). Conclusions. Off-hours and regular-hours percutaneous VA-ECMO implantation in cardiogenic shock of medical cause have similar results. Our results support well-designed 24/7 VA-ECMO implantation programs for cardiogenic shock patients.

## 1. Introduction

Venoarterial extracorporeal membrane oxygenation support (VA-ECMO) use has risen steeply over the last years for several cardiac conditions refractory to conventional measures (heart failure, pulmonary hypertension, myocarditis, post-cardiotomy shock, and intractable arrhythmias) [1], as a bridge to recovery or to cardiac transplantation [2]. However, the best timing to start VA-ECMO and the risk/benefit of this therapy remains unclear, as complications such as vascular damage, lower limb ischemia, and bleeding are common [3].

The “weekend effect” has been associated with an increased risk of in-hospital mortality in patients admitted during the night, weekends, and holidays [4]. This effect has been attributed to a reduction in the number of health-care professionals [5], greater severity of clinical presentation [6], and tiredness associated with long working hours [7]. In fact, admission to intensive care units during the weekend seems to be associated with in-hospital mortality [8]. In patients supported with VA-ECMO, previous studies have suggested that the prognosis may also worsen when this therapy is started during the weekend [9,10,11].

The aim of this study is to evaluate whether the results of off-hours and regular-hours peripheral VA-ECMO are comparable in an unselected population of patients with cardiogenic shock in a center with a “Shock Team” [12] and specific training of off-hours health-care professionals.

## 2. Materials and Methods

### 2.1. Study Population and Procedure

This observational, retrospective, single-center study included all patients who underwent peripheral VA-ECMO implantation for medical reasons at a high-volume tertiary hospital between 1 July 2013, and 30 September 2022, during regular-hours (weekdays 8:00 a.m. to 10:00 p.m.) and off-hours (weekdays 10:01 p.m. to 7:59 a.m., weekends, and holidays). The exclusion criteria were implantation in another center and periprocedural support (high-risk percutaneous coronary intervention or solid organ donation) (Figure 1). We registered the demographic characteristics, comorbidities, VA-ECMO indication, lactate level [13], and Society for Cardiovascular Angiography and Interventions (SCAI) stage [14] at the time of implantation, as well as complications and mortality (in-hospital and 90-day). All cannulations were performed percutaneously by an interventional cardiology team under ultrasound and/or fluoroscopy guidance. The cannulation team included one field nurse and one circulating nurse, and during regular hours two interventional cardiologists and during off hours only one interventional cardiologist. Initially, the membrane was primed immediately before implant; however, since January 2021, pre-primed membrane was available at the catheterization lab and was periodically recirculated. Implantation of an intra-aortic balloon pump or an Impella CP (Abiomed, Denver, CO, USA) for left ventricle venting was recommended in all cases after VA-ECMO implant, but the final decision was made by the on-call team on a case-by-case basis. Our protocol includes periodic training of the off-hours health professionals with bimonthly theoretical–practical sessions.

### 2.2. Definition and Outcomes

VA-ECMO indications due to medical reasons included refractory cardiogenic shock SCAI D-E (systolic blood pressure <90 mmHg, heart rate >120 beats per minute, lactate level >4 mmol/L, cardiac index <1.5 L/min/m^2^ despite treatment with norepinephrine >0.5 mcg/kg/min plus dobutamine or epinephrine), ventricular arrhythmia and arrhythmic storm, cardiopulmonary resuscitation, and acute pulmonary embolism. Complications included: acute kidney injury (increase in plasma creatinine value > 100% of baseline or the need of renal replacement therapy); major bleeding events (intracranial hemorrhage, bleeding requiring intervention to control, cardiac tamponade, or bleeding requiring transfusion >2 packed red blood cells in 8 h or >4 per day), hemolysis (plasma free hemoglobin levels >50 mg/dL), distal ischemia (requiring intervention, urgent decannulation, fasciotomy or amputation), ischemic or hemorrhagic stroke, and vascular lesion (hematoma with hemoglobin drop ≥1 g/dL, pseudoaneurysm, arterio-venous fistula, arterial thrombosis or deep venous thrombosis); and infection (all acquired during VA-ECMO support: ventilator-associated pneumonia, catheter-associated bacteriemia, or urinary tract infection; we excluded those acquired previous to VA-ECMO implant, e.g., aspiration pneumonia).

### 2.3. Statistical Analysis

Continuous variables were presented as median and interquartile range (IQR) or mean ± standard deviation when the normal distribution was observed, and were compared using the Kruskal–Wallis, Mann–Whitney U, and Student’s *t*-tests. Categorical variables were presented as number of patients and percentages, and were tested with the Pearson’s χ² test. The 90-day mortality was assessed using survival tables, Kaplan–Meier curves, and the log-rank test. All the tests were two-tailed. Statistical analyses were performed using SPSS Statistics 25.0 software (SPSS Inc., IBM, Chicago, IL, USA).

## 3. Results

### 3.1. Patients Characteristics

VA-ECMO was implanted in 147 patients with a median age of 56 years (IQR 49–64 years), and in 67 (45.6%) during off-hours. Appendix A shows the number of VA-ECMO implants made each year. Baseline characteristics are shown in Table 1 and were similar between regular-hours and off-hours groups. Patients were frequently male (76.2%) and had severe cardiogenic shock (median baseline lactate of 9.6 mmol/L [IQR 6.2–14.8 mmol/L] and >90% SCAI D or E stages). The most frequent indications were cardiogenic shock due to acute myocardial infarction (34%) and cardiopulmonary resuscitation (25%) (Figure 2).

In all but one case, femoro-femoral cannulation was performed. In most cases, a 23 French (F) extraction cannula (76%), 15 F return cannula (71%), and 6 F peripheral perfusion cannula (86%) were used. A left ventricular unloading device was implanted in 106 (72.1%) patients (intra-aortic balloon pump 96–65.3%, and Impella CP 20–13.6%). Compared with the regular-hours group, the off-hours group was more frequently treated with Impella CP (25.4% vs. 3.8%, *p* < 0.001) and a pre-primed membrane (27.9% vs. 13.2%, *p* = 0.039).

### 3.2. Clinical Outcomes

The mean time between the decision of VA-ECMO implant and the start of extracorporeal circulation (decision–implant time) was longer in the off-hours group (40.0 ± 30.7 min vs. 28.4 ± 13.1 min, *p* = 0.041). The mean duration of the implant procedure, from the first skin puncture during cannulation to the start of the extracorporeal circulation (implant time) was similar in both groups (28.4 ± 13.1 min vs. 26.9 ± 13.9 min, *p* = 0.518) (Figure 3). Appendix A shows implant time and decision–implant time according to year of implant.

The median duration of VA-ECMO support was 4 days (IQR 2–6 days) and the length of hospital stay was 32 days (IQR 17–64 days), with no relevant differences between regular-hours and off-hours groups. Decannulation was achieved in 50 regular-hours patients (62.5%) and 40 off-hours patients (59.7%), *p* = 0.729. In-hospital mortality was similar in off-hours (37–55.2%) and regular hours (45–56.3%), *p* = 0.901 (Table 2). In addition, no relevant differences were found in mortality after the 90-day follow-up (log-rank *p* = 0.963) (Figure 4). The results were similar irrespective of VA-ECMO indications (Appendix A) and type of left ventricular unloading device (Appendix A). Most frequent causes of death were multiorgan failure (30–34.1%), severe brain injury (21–23.9%), uncontrollable bleeding (9–10.2%), intracranial hemorrhage (6–6.8%), sepsis (5–5.7%), arrhythmic storm (4–4.5%), and end-stage heart failure (4–4.5%); without differences between regular-hours and off-hours groups (*p* = 0.76).

The most common complications were acute kidney injury (55–37.4%, with need of renal replacement therapy in 15 cases), major bleeding event (51–34.7%), severe lower limb ischemia (28–19.0%), vascular injuries (25–17.9%), infections (23–15.6%), and stroke (9–6.1%). Despite frequent use of a left ventricle unloading device, left ventricle overdistention occurred in 17 patients (11.6%). Complications were similar in both groups, except of a higher incidence of stroke off-hours (10.4% vs. 2.5%, *p* = 0.045) (Table 2).

## 4. Discussion

In our study, performed in a center with a “Shock Team” and specific training of off-hours health-care professionals, regular-hours and off-hours VA-ECMO implantation had similar results in terms of in-hospital and 90-day mortality, weaning success, and length of hospital stay. Complications related to VA-ECMO therapy were also comparable.

The “weekend effect” has been described in different conditions [4] and even meta-analyses have reported an association between off-hours admission and worse outcomes in heart failure, cardiorespiratory arrest, and acute coronary syndromes [15,16]. A few previous studies have suggested unfavorable VA-ECMO results in cardiopulmonary resuscitation patients cannulated off-hours [11,17,18]. This disadvantage might be due to a reduction in the number of health-care professionals [5], greater severity of clinical presentation [6], and tiredness associated with long working hours [7]. However, longer delays might also play a role. In our study, although the cannulation time was comparable in the two groups, the decision–implantation time was higher off-hours than regular-hours. Nevertheless, it is unclear if a 30-min delay to start VA-ECMO support has a prognostic impact [19].

In a single-center study that included 200 VA-ECMO implants in the setting of cardiorespiratory arrest, survival results were worse during the weekend [11]. As in our case, the time from the cardiac arrest to the start of VA-ECMO support was longer off-hours (47 min vs. 31 min) [11]. The relevance of this delay is probably much more important in cardiopulmonary resuscitation than in cardiogenic shock. In fact, a study that included 250 patients with VA-ECMO implantation in cardiogenic shock, found similar results to our series, as cannulation outside working hours was not associated with increased mortality, duration of ECMO support, or longer intensive care unit stay [10]. The only negative association with off-hours in this study were vascular complications [10], something we did not observe, probably due to the generalization in the implantation of peripheral perfusion cannula. In addition, in a pediatric population, off-hours VA-ECMO implantation is also not associated with a higher rate of complications or mortality [20,21], even in the setting of cardiopulmonary resuscitation [17].

Compared to patients on regular hours, off-hours patients had a higher incidence of stroke; the explanation of this possible association remains unclear. VA-ECMO support may increase the risk of stroke due to several factors such as need for systemic anticoagulation, possibility of aortic root or left ventricle thrombosis, systemic inflammation with capillary fragility, or hemolysis [22]. In addition, the concomitant use of Impella, more frequent in off-hours group, may increase the risk of thrombotic and hemorrhagic complications [23], although most cerebrovascular events occurred in patients not supported with Impella. A previous VA-ECMO series also reported a high incidence of stroke (5.8%), with hyperlactatemia being the only independent predictor [24].

The main interest of our work lies in the fact that we included patients with cardiogenic shock due to multiple medical (non-surgical) reasons, beyond VA-ECMO implant in the context of cardiopulmonary resuscitation, in which the cannulation was performed in all cases by an interventional cardiologist team. In our series, off-hours versus regular-hours comparable outcomes are probably thanks to off-hours staff training and the on-call Shock Team/interventional cardiologist availability to ensure a correct patient selection and VA-ECMO cannulation procedure.

Our study also has some limitations. As it is a retrospective observational revision, we could have underestimated differences in the severity of the clinical condition between groups, although the absence of relevant differences at baseline, including lactate values and SCAI stage, suggests that our two groups of patients are comparable. In addition, this is a single-center study, in a high-volume academic hospital, so our data results may not be generalizable to other centers with less experience in VA-ECMO implantation. Over 90% of VA-ECMO implants in our center are guided by ultrasound plus fluoroscopy and less than 10% only by fluoroscopy; unfortunately, we have not collected data about guidance of the implant with fluoroscopy or ultrasound in each patient. Finally, the sample size is smaller than some of the previously published series, but we have not included VA-ECMO implantation in cardiac surgery-related cardiogenic shock, a common cause in previous cohorts.

## 5. Conclusions

Off-hours and regular-hours percutaneous VA-ECMO implantation in cardiogenic shock of medical cause have similar results. Our results support well-designed 24/7 VA-ECMO implantation programs for cardiogenic shock patients that should include specific training of all health-care professionals that might be involved in off-hours VA-ECMO implantation and management.

## Figures and Tables

**Figure 1 jcm-12-01875-f001:**
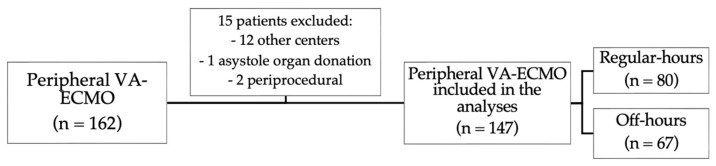
Flow chart. VA-ECMO: venoarterial extracorporeal membrane oxygenation.

**Figure 2 jcm-12-01875-f002:**
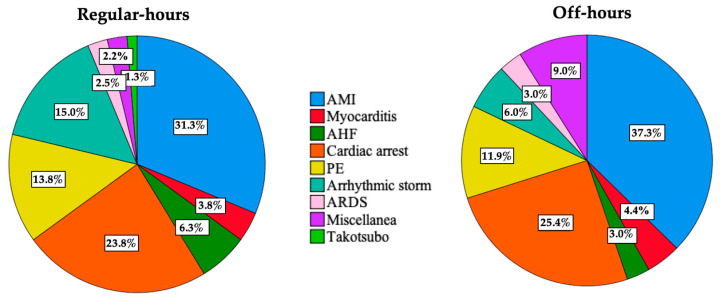
Reasons for venoarterial extracorporeal membrane oxygenation indication. No significant differences were found between regular-hours and off-hours groups (*p* = 0.467). AMI: acute myocardial infarction. AHF: acute heart failure. PE: pulmonary embolism. ARDS: acute respiratory distress syndrome.

**Figure 3 jcm-12-01875-f003:**
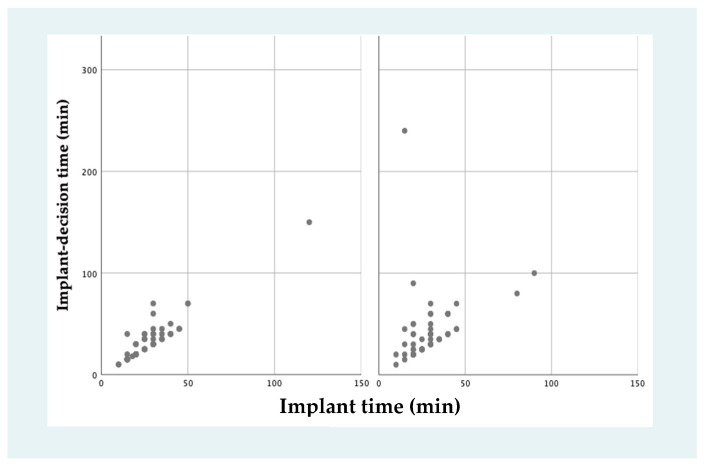
Venoarterial extracorporeal membrane oxygenation implant times. Time between venoarterial extracorporeal membrane oxygenation implant decision and the end of the implant procedure was slightly higher in off-hours group (40.0 ± 30.7 min vs. 28.4 ± 13.1 min, *p* = 0.041), without differences in implant procedure times.

**Figure 4 jcm-12-01875-f004:**
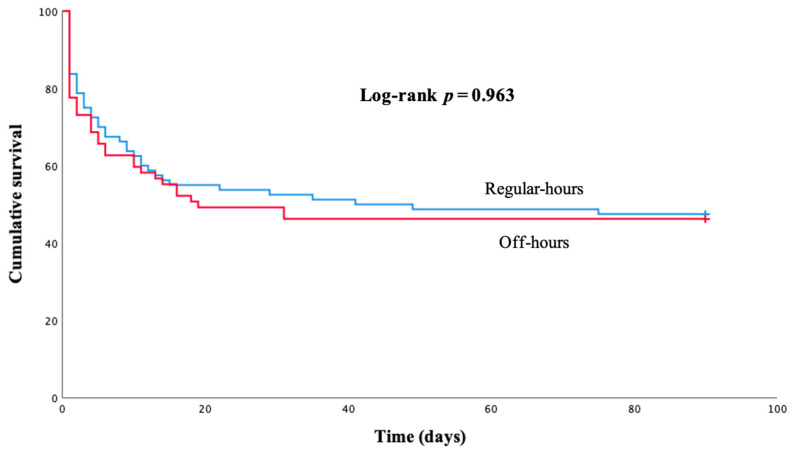
Kaplan–Meier survival curve for 90-day follow-up according to the moment of venoarterial extracorporeal membrane oxygenation implantation. Log-rank *p* = 0.963.

**Table 1 jcm-12-01875-t001:** Baseline characteristics according to the moment of venoarterial extracorporeal membrane oxygenation implantation.

	Total (n = 147)	Regular-Hours (n = 80)	Off-Hours (n = 67)	*p*
Age, years	56 (49–64)	55 (49–63)	58 (49–65)	0.328
Male sex, n (%)	112 (76.2)	62 (77.5)	50 (74.6)	0.684
BMI ^1^, kg/m^2^	27.7 (25–31.1)	27.7 (25.2–31.4)	27.7 (24.8–29.9)	0.562
Hypertension, n (%)	62 (42.2)	35 (43.8)	27 (40.3)	0.673
Diabetes mellitus, n (%)	33 (22.4)	20 (25)	13 (19.4)	0.418
Dyslipidemia, n (%)	64 (43.5)	32 (40.0)	32 (47.8)	0.345
Active smoker, n (%)	49 (33.3)	27 (33.8)	22 (32.8)	0.988
Ischemic heart disease, n (%)	26 (17.7)	11 (13.8)	15 (22.4)	0.172
Atrial fibrillation, n (%)	16 (10.9)	10 (12.5)	6 (9.0)	0.492
Peripheral artery disease, n (%)	15 (10.2)	10 (12.5)	5 (7.5)	0.315
Creatinine, mg/dL	1.3 (1.1–1.8)	1.3 (1.1–1.7)	1.3 (1.1–1.8)	0.949
Lactate, mmol/L	9.6 (6.2–14.8)	8.9 (4.1–14)	10.4 (7.5–14.9)	0.122
LVEF ^2^, %	20 (10–40)	23 (10–45)	20 (10–30)	0.521
SCAI ^3^ D or E, n (%)	136 (92.5)	75 (93.8)	61 (91.0)	0.272

^1^ BMI: body mass index. ^2^ LVEF: left ventricular ejection fraction. ^3^ Society for Cardiovascular Angiography and Interventions shock stage classification.

**Table 2 jcm-12-01875-t002:** Clinical outcomes according to the moment of venoarterial extracorporeal membrane oxygenation implantation.

	Total (n = 147)	Regular-Hours (n = 80)	Off-Hours (n = 67)	*p*
In-hospital mortality, n (%)	82 (55.8)	45 (56.3)	37 (55.2)	0.901
Length of VA-ECMO support, days	4 (2–6)	4 (2–6)	4 (2–5)	0.46
Successful decannulation, n (%)	90 (60.4)	50 (62.5)	40 (59.7)	0.729
Length of hospital stay, days	32 (16.5–64)	32 (18–63)	31 (16–65.8)	0.979
Complications				
Acute kidney injury, n (%)	55 (37.4)	29 (36.3)	26 (38.8)	0.696
RRT ^1^ need, n (%)	15 (10.2)	8 (10)	7 (10.4)	0.929
Major bleeding, n (%)	51 (34.7)	25 (31.3)	26 (38.8)	0.338
Distal ischemia, n (%)	28 (19)	15 (18.8)	13 (19.4)	0.92
Vascular injury, n (%)	25 (17)	12 (15)	13 (19.4)	0.479
Stroke, n (%)	9 (6.1)	2 (2.5)	7 (10.4)	0.045
Infection, n (%)	23 (15.6)	14 (17.5)	9 (13.4)	0.499
LV ^2^ overdistention, n (%)	17 (11.6)	10 (12.5)	7 (10.5)	0.698

^1^ RRT: renal replacement therapy. ^2^ LV: left ventricle.

## Data Availability

The datasets used and analyzed in the current study are made available from the corresponding author on reasonable request.

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
