# Peer review of "Off-Hours versus Regular-Hours Implantation of Peripheral Venoarterial Extracorporeal Membrane Oxygenation in Patients with Cardiogenic Shock"

_jcm, 2023, doi:10.3390/jcm12051875_

Round 1
Reviewer 1 Report
The present analysis, "Off-hours versus regular-hours implantation of peripheral venoarterial extracorporeal membrane oxygenation in patients with cardiogenic shock," presents an interesting hypothesis that is certainly frequently asked and cannot be clearly answered. The nicely presented results show no difference between the two groups of regular- and off-hours patients except for the stroke rate. This again is not a trend towards a higher incidence (line 215) - it is significant. Why, of course, cannot be clarified here, although it is a relatively large group of patients. Perhaps this can be worked out a little more clearly here. Nevertheless, the results are discussed in detail and I think the article is well worth. A multicenter analysis could be performed here.
Author Response
Reviewer 1.
The present analysis, "Off-hours versus regular-hours implantation of peripheral venoarterial extracorporeal membrane oxygenation in patients with cardiogenic shock," presents an interesting hypothesis that is certainly frequently asked and cannot be clearly answered. The nicely presented results show no difference between the two groups of regular- and off-hours patients except for the stroke rate.
We would like to thank the comments from reviewer 1 that have helped us to improve our manuscript.
This again is not a trend towards a higher incidence (line 215) - it is significant. Why, of course, cannot be clarified here, although it is a relatively large group of patients. Perhaps this can be worked out a little more clearly here. Nevertheless, the results are discussed in detail and I think the article is well worth. A multicenter analysis could be performed here.
This has been clarified. Our data suggest an association between off-hour implantation and stroke (10.4% vs 2.5%), although, as we have studied several associations, the relatively high p value (0.045) should be interpreted with caution, and a random association in a retrospective study is also a possibility. We agree that a future multicenter study would be of great value in finding some explanation for this and other questions.
Reviewer 2 Report
Thanks for allow me to review this interesting research work and I take this opportunity to congratulate the authors for their wonderful job.
After analyzing the paper, I would like to make some recommendations, with the aim of trying to improve the quality of this manuscript.
2. Materials and Methods
2.1. Study Population and Procedure
I recommend the authors to expand on the description of the team of canulators, how many of them make up the team, is it made up of the same people between regular and off-hours?...
2.2 Definition and outcomes.
I have missed that differential hypoxia and left ventricular overdistention have not been documented among the complications.
It is relevant to report these complications, especially in the off-hours setting, as in cases and under certain circumstances where these complications occur outside of regular hours, it would (but not always) imply performing other cannulation-related procedures, such as changing cannulation strategy or adding a second cardiac assist device to unload the left ventricle. Although the frequency of use of Impella CP is described in the results section.
3. Results
3.1. Patients Characteristics.
The patient population is described, indicating that the majority were in severe cardiogenic shock, although only lactate and SCAI stage variables are included. Patients were frequently male (76.2%) and had severe cardiogenic shock (median baseline lactate of 9.6 mmol/L [IQR 6.2-14.8 mmol/L] and > 90% SCAI D or E stages).
Although serum lactate level is a good prognostic marker, it is not a reliable marker to discern or differentiate between different severity between SCAI stages C and D for instance, therefore, I recommend including the mean cardiac index, in order to make a more objective description.
Regarding complications. The incidence of strokes in off-hours patients is relevant, as has a statistical significance difference (0.045).
I’m wondering how authors differentiated between ischemic events related to thrombus (thromboembolic events as stroke definition) or secondary to cerebral ischemia due to hemodynamic instability and impaired cerebral perfusion from ECLS?
4. Discussion.
Re-assess the potential role of other likely complications related to the trend toward high incidence of “stroke” in off-hours patients, as differential hypoxia.
Regarding the case/year experience of VA ECMO in the study, I suggest more modesty to the authors. Most centers with high volume recognition in Europe report between 30-50 cases of ECMO VA per year. In fact, some centers in Germany, England, and France report more than 100 cases/year. Even adding post-cardiotomy VA ECMO cases, this analysis might be biased, as I assume that the cannulation team (Shock Team) is not the same.
5. Conclusions.
Once again, if the authors want to support their organizational model for a "Shock Team" with the results presented, the structure of the team and its particularity should be better described.
Author Response
Thanks for allow me to review this interesting research work and I take this opportunity to congratulate the authors for their wonderful job.
After analyzing the paper, I would like to make some recommendations, with the aim of trying to improve the quality of this manuscript.
We would like to thank reviewer by the comments that have helped us to improve our manuscript.
2. Materials and Methods
2.1. Study Population and Procedure
I recommend the authors to expand on the description of the team of canulators, how many of them make up the team, is it made up of the same people between regular and off-hours?...
In section 2.1. Study Population and Procedure, we report the composition of the cannulation team: “During regular-hours, the team included one field nurse and one circulating nurse, as well as two interventional cardiologists, all of them physically present in the hospital. During off-hours, the team also included one field nurse and one circulating nurse, but only one interventional cardiologist, all of them available on call at home.”
2.2 Definition and outcomes.
I have missed that differential hypoxia and left ventricular overdistention have not been documented among the complications.
It is relevant to report these complications, especially in the off-hours setting, as in cases and under certain circumstances where these complications occur outside of regular hours, it would (but not always) imply performing other cannulation-related procedures, such as changing cannulation strategy or adding a second cardiac assist device to unload the left ventricle. Although the frequency of use of Impella CP is described in the results section.
Left ventricular overdistention occurred in 17 patients (11.6%), 7 off-hours, and 10 regular-hours (p = 0.698). We have added this in the results.
3. Results
3.1. Patients Characteristics.
The patient population is described, indicating that the majority were in severe cardiogenic shock, although only lactate and SCAI stage variables are included. Patients were frequently male (76.2%) and had severe cardiogenic shock (median baseline lactate of 9.6 mmol/L [IQR 6.2-14.8 mmol/L] and > 90% SCAI D or E stages).
Although serum lactate level is a good prognostic marker, it is not a reliable marker to discern or differentiate between different severity between SCAI stages C and D for instance, therefore, I recommend including the mean cardiac index, in order to make a more objective description.
We agree that hemodynamic parameters such as mean cardiac index would improve the characterization of the population, but unfortunately, we have not recorded these data.
Regarding complications. The incidence of strokes in off-hours patients is relevant, as has a statistical significance difference (0.045).
This has been clarified. Our data suggest an association between off-hour implantation and stroke (10.4% vs 2.5%), although, as we have studied several associations, the relatively high p value (0.045) should be interpreted with caution, and a random association in a retrospective study is also a possibility.
I’m wondering how authors differentiated between ischemic events related to thrombus (thromboembolic events as stroke definition) or secondary to cerebral ischemia due to hemodynamic instability and impaired cerebral perfusion from ECLS?
We have not enough data to differentiate between thromboembolic events and those secondary to cerebral ischemia in the context of impaired cerebral perfusion.
4. Discussion.
Re-assess the potential role of other likely complications related to the trend toward high incidence of “stroke” in off-hours patients, as differential hypoxia.
We agree that other complications such as differential hypoxia could contribute to the higher incidence of stroke in off-hours patients but unfortunately, we have not recorded it as a complication. There were no differences regarding the need to switch to a VAV-ECMO configuration between regular-hours and off-hours groups: a veno-arterial-venous ECMO configuration (VAV-ECMO) was necessary in 5 patients (3.4%), 2 of them in regular-hours group and other 3 in off-hours group (p = 0.824). A veno-venous-arterial ECMO configuration (VVA-ECMO) was used in 1 patient of regular-hours group due to suction problems.
Regarding the case/year experience of VA ECMO in the study, I suggest more modesty to the authors. Most centers with high volume recognition in Europe report between 30-50 cases of ECMO VA per year. In fact, some centers in Germany, England, and France report more than 100 cases/year. Even adding post-cardiotomy VA ECMO cases, this analysis might be biased, as I assume that the cannulation team (Shock Team) is not the same.
Sample size is included as a limitation. However, we think that the inclusion of only medical causes of cardiogenic shock and only percutaneous cannulated VA-ECMO by interventional cardiologists should be considered an added value of our cohort.
5. Conclusions.
Once again, if the authors want to support their organizational model for a "Shock Team" with the results presented, the structure of the team and its particularity should be better described.
Our organizational model for a “Shock Team” has been previously reported, as you can find in reference 12.
Reviewer 3 Report
Gómez-Sánchez and colleagues conducted a very interesting study. Some points need to be addressed:
Definition of on- vs. off-hours: Do the results differ by narrowing the on hours (e.g. 08:00-16:00)?
Methods: Please provide information about how many cases very made each year (2013-2022)
65: The authors report: “All cannulations were performed percutaneously by an interventional cardiology team under ultrasound and/or fluoroscopy guidance” – Does the authors observe different results according to the one of these approaches in terms of duration and/or outcome?
132: Please describe the decision making process for LV unloading (because not all patients received this strategy) and at which time point it was performed (directly after VA-ECMO implantation?)
137: “end of cannulation” = ECMO running? Please indicate/specify
139: “duration of the implant procedure” – please define/specify
137-139: “mean time between the decision of VA-ECMO implant and the end of cannulation” “duration of the implant procedure” – Please provide information specifically for each year
The reasons for VA-ECMO implantation were diverse – The authors should provide a supplementary table in which they describe mortality rates for each indication (as shown in Figure 2) separately and compare them between on- and off-hours.
Any explanations why patients during off-hours were more likely to receive Impella CP support?
Clinical outcomes: The authors should provide specific information about the cause of death.
The authors should provide (supplementary file) results (mortality) comparing a) unloading vs. no unloading b) IABP vs. Impella
Round 2
Reviewer 3 Report
Thank you for providing the revised version.